# Rapid labelling and covalent inhibition of intracellular native proteins using ligand-directed *N*-acyl-*N*-alkyl sulfonamide

Tomonori Tamura [1], Tsuyoshi Ueda[1], Taiki Goto[1], Taku Tsukidate[1], Yonatan Shapira[2], Yuki Nishikawa[1], Alma Fujisawa[1] & Itaru Hamachi[1,3]

Selective modification of native proteins in live cells is one of the central challenges in recent chemical biology. As a unique bioorthogonal approach, ligand-directed chemistry recently emerged, but the slow kinetics limits its scope. Here we successfully overcome this obstacle using *N*-acyl-*N*-alkyl sulfonamide as a reactive group. Quantitative kinetic analyses reveal that ligand-directed *N*-acyl-*N*-alkyl sulfonamide chemistry allows for rapid modification of a lysine residue proximal to the ligand binding site of a target protein, with a rate constant of $\sim 10^4 \, M^{-1} s^{-1}$, comparable to the fastest bioorthogonal chemistry. Despite some off-target reactions, this method can selectively label both intracellular and membrane-bound endogenous proteins. Moreover, the unique reactivity of *N*-acyl-*N*-alkyl sulfonamide enables the rational design of a lysine-targeted covalent inhibitor that shows durable suppression of the activity of Hsp90 in cancer cells. This work provides possibilities to extend the covalent inhibition approach that is currently being reassessed in drug discovery.

[1] Department of Synthetic Chemistry and Biological Chemistry, Graduate School of Engineering, Kyoto University, Katsura, Nishikyo-ku, Kyoto 615-8510, Japan. [2] Feinberg Graduate School, Weizmann Institute of Science, Rehovot 7610001, Israel. [3] CREST (Core Research for Evolutional Science and Technology, JST), Sanbancho, Chiyodaku, Tokyo 102-0075, Japan. Correspondence and requests for materials should be addressed to I.H. (email: ihamachi@sbchem.kyoto-u.ac.jp)

Chemical modification of target proteins with synthetic molecules is a powerful methodology for the generation of a myriad of engineered proteins, such as antibody–drug conjugates and for the detailed study of protein function and dynamics in living cells. Over the past two decades, bioorthogonal methods, including chemoselective reactions with non-canonical amino acids bearing bioorthogonal functional groups, protein/peptide-tag-based methods, self-labelling enzyme tags and enzyme-mediated modifications have proven to be valuable for efficient and selective protein labelling[1–9]. Because these reactions are usually conducted in dilute and/or multi-molecular crowding live cell conditions, unlike conventional chemical reactions in a flask, fast and highly chemoselective reactions are desired[2,5,6]. Remarkable advances have been now achieved in both these parameters[10,11], providing researchers with useful tools for the selective chemical modification and engineering of protein molecules, even under live cell conditions. Most bioorthogonal protein modification strategies typically require two steps: (i) the incorporation of a bioorthogonal reaction handle, such as a noncanonical amino acid or an enzyme domain or its substrate peptide, into a protein of interest (POI) using genetic engineering protocols, followed by (ii) a specific (bioorthogonal) reaction to attach a functional molecule, such as a drug, fluorophore, polymer or detection tag (Fig. 1a)[5]. Although powerful, this two-step method inevitably involves genetic manipulation, making it impossible to modify endogenous (i.e. naturally occurring) proteins in their native environments.

As an alternative to the bioorthogonal protein labelling approaches, we have developed a chemical strategy based on an affinity-driven reaction involving protein–ligand interactions, termed ligand-directed (LD) chemistry (using systems bearing tosyl (LDT)[12–14], alkyloxyacyl imidazole (LDAI)[15–17] and dibromophenylbenzoate (LDBB)[18] reactive moieties). LD chemistry is a simple one-step procedure employing a labelling reagent in which a ligand for a POI and an appropriate functional molecule are connected with a cleavable electrophilic group. The reagent selectively binds to the POI through a specific protein–ligand interaction, facilitating the transfer of a functional molecule to a nucleophilic amino acid residue close to the ligand binding site via covalent bond formation[4,12]. It is thus considered that the protein

selectivity and the labelling kinetics of LD chemistry may largely rely on the recognition-driven proximity effect. However, no quantitative kinetic analyses have been performed yet, and therefore LD chemistry cannot be clearly evaluated, compared with conventional bioorthogonal methods, as a complimentary tool for protein chemical modification.

One of the most unique characteristics of LD chemistry is that it can target canonical amino acids of endogenous proteins with sufficient selectivity even in intact living systems without genetic disturbance[12,19]. This advantage let us envision that LD chemistry could be applied for the functional regulation of native proteins, as well as a complement to bioorthogonal chemistry[20,21]. For example, the rational design of a LD reagent that is able to covalently attach a ligand (instead of a probe) to a protein should be able to produce targeted covalent inhibitors (TCIs). The development of unique TCIs may lead to potent covalent drugs, which have undergone a recent resurgence in drug discovery research because of their durability[22,23]. To date, most of the recently developed TCIs target cysteine[22–26], and expansion of the available targetable amino acids remains a great challenge[27,28]. Given that our LD reagents have been shown to react with several canonical amino acids (e.g. Lys, His, Ser, Tyr and Glu) other than Cys[12,13,15,18], our efforts to develop a variety of LD chemistry may substantially contribute to extending chemical warheads that can be used for TCIs (and covalent drugs)[28].

Herein, we describe a LD chemistry using $N$-acyl-$N$-alkyl sulfonamide derivatives (LDNASA) as the electrophilic reactive group (Fig. 1b). Detailed kinetic analyses of in vitro protein labelling are conducted and compared with other LD reagents, revealing that the reaction rate of the labelling process ($k_L$) of LDNASA is the fastest of all the LD chemistry reagents (LDT, LDAI and LDBB). Notably, the second-order reaction rate constant ($k_L/K_d$) of LDNASA-mediated protein labelling reaches $\sim 10^4\,M^{-1}\,s^{-1}$, which is comparable to enzymatic labelling and the fastest bioorthogonal protein bioconjugations, such as inverse electron-demand Diels-Alder (IEDDA) cycloadditions. The excellent kinetics and bioorthogonality of LDNASA chemistry allow for the selective and efficient labelling of intracellular endogenous proteins, as well as a membrane-protein, in live cell

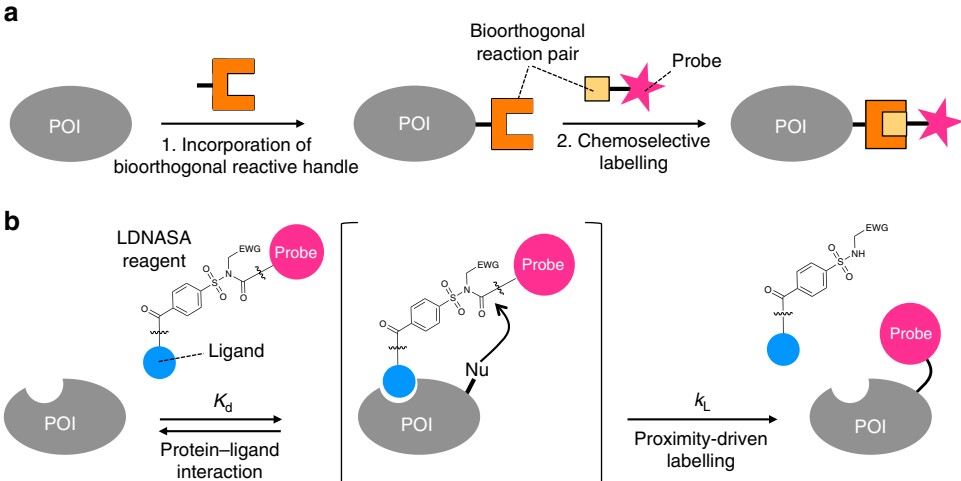

**Fig. 1** Two distinct approaches for bioorthogonal protein labelling. **a** Schematic illustration of protein labelling through bioorthogonal chemistry. The first step involves the genetic incorporation of a bioorthogonal reactive handle into a protein of interest (POI) in cells. In a second step, the reactive handle chemoselectively reacts with a designed synthetic probe. **b** Schematic illustration of the basic principle of ligand-directed $N$-acyl-$N$-alkyl sulfonamide (LDNASA) chemistry. The reagent binds to a POI through a specific protein–ligand interaction, driving a chemical reaction between the reactive group and a natural nucleophilic amino acid located on the protein surface, through the proximity effect. EWG, electron-withdrawing group; $K_d$, dissociation constant; $k_L$, first-order rate constant for the labelling process; Nu, nucleophilic amino acid

habitats. More interestingly, we demonstrate that the NASA reactive group is a promising warhead for a covalent inhibitor, which can irreversibly suppress the activity of molecular chaperone protein Hsp90 endogenously expressed in living cancer cells.

## Results

**Design and characterisation of LDNASA reagents**. While the LD chemistries reported by our group (LDT, LDAI and LDBB) have proved to be applicable to selective protein labelling both in vitro and in living cells, they still suffer from sluggish reaction rates, requiring hours to a day to achieve acceptable levels of labelled products[14,15,18,19]. This motivated us to further explore an electrophilic cleavable linker amenable to the LD strategy. We here focused on NASA, which has been widely used as Kenner's safety-catch linker in solid-phase peptide synthesis[29]. Recently, our group found that NASA derivatives can be used as acyl donors for catalyst-mediated protein labelling in vitro and in crude biological environments, such as live-cell surfaces and brain tissues[30]. This study revealed that NASA is sufficiently stable in neutral aqueous buffer and cell lysate conditions, and is less susceptible to enzymatic degradation because of its unique structure not found in nature. We thus reasoned that NASA may have potential as a promising cleavable linker in LD chemistry for bioorthogonal protein acylation. It has been reported that the electrophilicity of the N-acyl group is distinctly dependent on the electron withdrawing properties of the N-alkyl group[29,31]. To identify the optimal N-alkyl group of NASA for LD chemistry, cyano, 4-nitrophenyl, or 2,4-dinitrophenyl substituents were selected as the electron-withdrawing groups (EWGs) on the basis of our previous study[30]. We first conducted in vitro experiments

using FKBP12 as a model target protein. We thus designed NASA reagents 1–3 (Fig. 2a), in which a synthetic ligand for FKBP12 (SLF, reported $K_d = 20$ nM)[32] and a biotin (Bt) affinity-tag are connected with a NASA group. The syntheses were carried out as shown in Supplementary Methods. All of the final compounds were well characterised by NMR and high-resolution mass spectroscopy.

FKBP12 labelling was conducted in test tubes by incubating each NASA reagent (1–3) (10 μM) with purified recombinant (wild-type) FKBP12[13] (5 μM, $M_w$: 11 914 Da) at 37 °C in a buffer solution (pH 7.2). The labelling reactions were monitored by matrix-assisted desorption/ionisation time-of-flight mass spectrometry (MALDI-TOF MS). In all cases, the molecular mass corresponding to Bt-labelled FKBP12 ($M_w$: 12 140 Da) was detected as shown in Fig. 2b and Supplementary Figure 1. Rapid and efficient FKBP12 labelling was observed in the case of 1 (98% yield within 15 min), whereas 2 and 3 gave lower yields (22% for 2, 17% for 3) even after a 2-h incubation. The initial rate of the reaction with 1 (84 μM min⁻¹) was 49- and 105-fold faster than 2 (1.7 μM min⁻¹) and 3 (0.8 μM min⁻¹), respectively (Fig. 2c). The labelling by 1 was completely abolished in the presence of rapamycin[33] (a competitive ligand), indicating that the labelling reaction was efficiently driven by an affinity-mediated proximity effect (Fig. 2b, c). This rapid and efficient labelling using a NASA containing a cyanomethyl group was also observed when targeting Escherichia coli dihydrofolate reductase (eDHFR) using reagent 4 containing trimethoprim (TMP) as a ligand (15 min, 80%) (Fig. 3a and Supplementary Figure 2)[18,34]. The results clearly showed that the cyano group is the optimal N-alkyl group in LDNASA reagents for efficient protein labelling. The intrinsic reactivity of the reagents was evaluated by the hydrolysis rate in

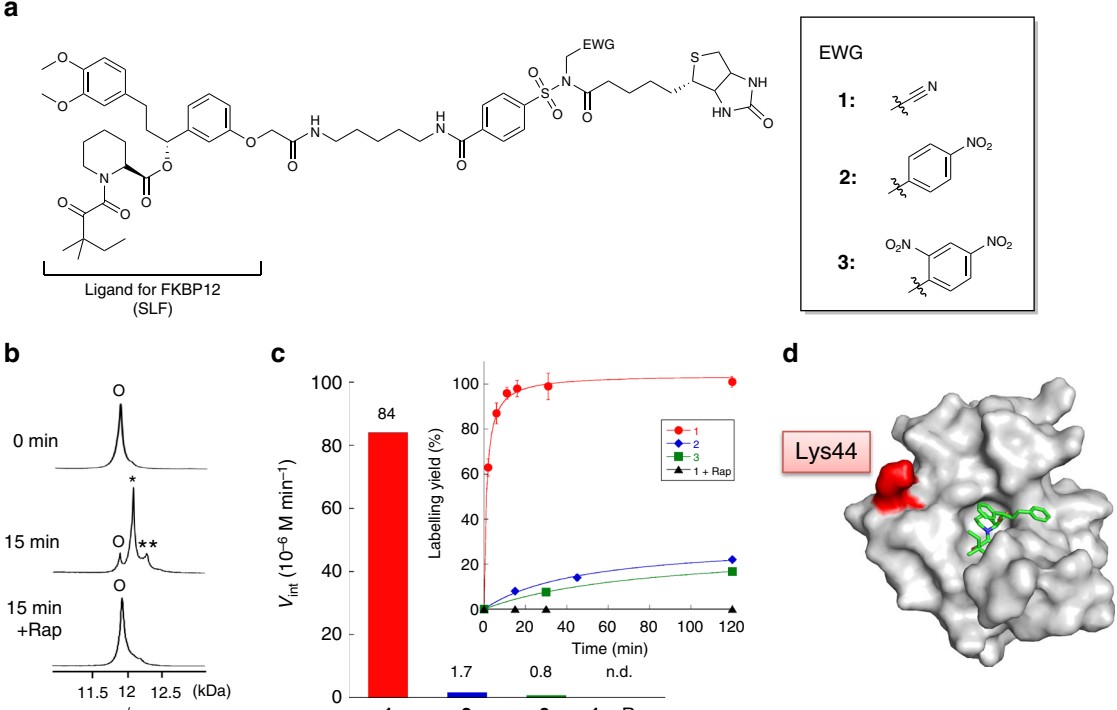

**Fig. 2** In vitro FKBP12 labelling with LDNASA reagents 1–3. **a** Molecular structures of LDNASA reagents 1–3. **b** MALDI-TOF mass analysis of FKBP12 labelling by 1 in the absence or presence of rapamycin (Rap). Reaction conditions: 5 μM FKBP12, 10 μM 1, 100 μM Rap, 50 mM HEPES buffer, pH 7.2, 37 °C. o, native FKBP12 ($M_w$: 11 914); *, single-labelled FKBP12 ($M_w$: 12 140); **, double-labelled FKBP12 ($M_w$: 12 366). **c** Initial rates (M min⁻¹) of FKBP12 labelling by 1–3. The initial rates were estimated from the time courses of the reaction yields (inset). n.d., not detected. **d** The crystal structure of the FKBP12-SLF complex (PDB:1FKG). Lys44, the major labelling site with 1, is coloured in red. The SLF ligand is shown as a green stick. Lys34 was also identified as the second (minor) labelling site (see Supplementary Figure 5)

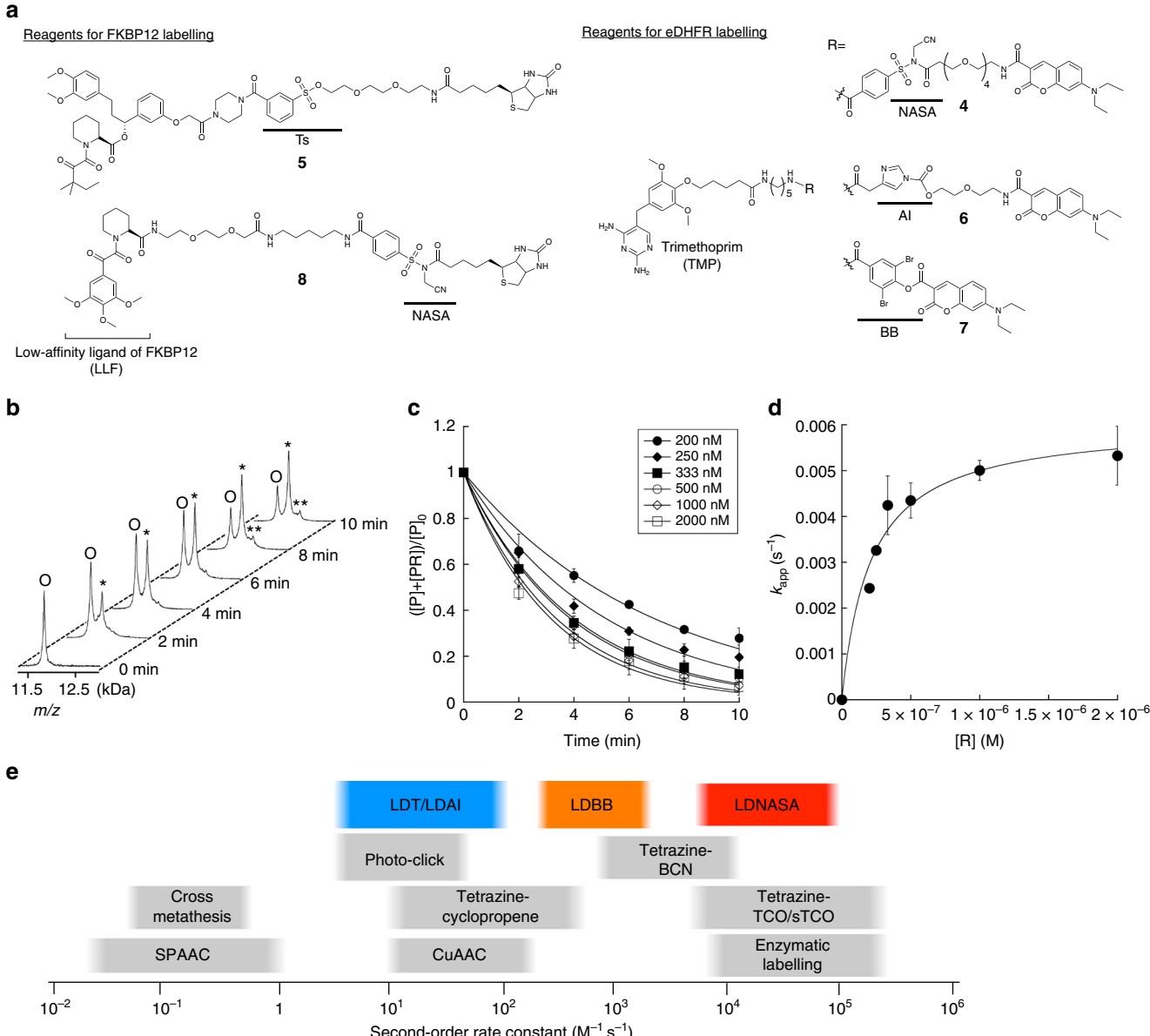

**Fig. 3** Kinetic analysis of protein labelling with various LD reagents. **a** Molecular structures of LD reagents. NASA, *N*-acyl-*N*-alkyl sulfonamide; Ts, tosyl; AI, alkyloxyacyl imidazole; BB, dibromophenyl benzoate. **b** MALDI-TOF MS analyses of labelling process of FKBP12 with **1**. Purified FKBP12 (100 nM) was incubated with **1** (200 nM) in HEPES buffer (50 mM, pH 7.2) at 37 °C. O, native FKBP12 ($M_w$: 11 914); *, single-labelled FKBP12 ($M_w$: 12 140); **, double-labelled FKBP12 ($M_w$: 12 366). **c** Time course of the depletion of native (non-labelled) FKBP12 during the labelling reaction with various concentrations of **1**. Purified FKBP12 (100 nM) was incubated with reagents (200–2000 nM) in HEPES buffer (50 mM, pH 7.2) at 37 °C. The reaction was monitored by MALDI-TOF MS. The pseudo-first order reaction rates ($k_{app}$) were obtained by fitting the data to Eq. (2). **d** The dependence of $k_{app}$ upon the concentration of **1**. Kinetic parameters were obtained by fitting this data to Eq. (3). **e** Second-order rate constants of representative bioorthogonal reactions and LD chemistries for protein modification. Here, the rate constants for LD chemistries indicate the range of values when the affinity ($K_d$) of the ligand is $10^{-7}$–$10^{-6}$ M. The rate constants for the bioorthogonal reactions were obtained from refs [5,10]. Error bars represent s.d., $n = 3$

aqueous buffer. The half-life of **1** was determined to be 43 h from HPLC analysis, whereas only 5% of **2** and **3** were hydrolysed even after incubation for 48 h at 37 °C (Supplementary Figure 3). This suggested that the rapid labelling kinetics of *N*-acyl-*N*-cyano-methyl sulfonamide can be mainly attributed to the higher electrophilicity of *N*-acyl-*N*-cyanomethyl sulfonamide compared with *N*-acyl-*N*-(mono or di) nitrobenzyl sulfonamide.

The labelling sites on FKBP12 after reaction with **1** were identified by conventional peptide mapping analysis. FKBP12 labelled by **1** was digested with trypsin and the resultant peptide fragments were analysed by HPLC followed by tandem mass

spectrometry. As shown in Fig. 2d, Supplementary Figures 4 and 5, Lys44 located near the ligand-binding pocket (11.3 Å from the bound SLF ligand) was predominantly modified with biotin by **1**. We also identified the labelling site of eDHFR to be specific to Lys32 when **4** was used (Supplementary Figure 6). These results clearly demonstrate that NASA reagents prefer the ε-amino group of lysine side chains, and form a chemically stable amide bond.

**Kinetic analysis of ligand-directed chemistry.** With the optimal NASA reactive group in hand, we next investigated the reaction kinetics of LD chemistry in in vitro protein labelling. The

labelling reaction of LD chemistry follows Eq. (1):

$$P + R \overset{K_d}{\leftrightarrows} PR \overset{k_L}{\longrightarrow} P^* \qquad (1)$$

where P is a native (non-labelled) protein, R is the reagent, PR is the native protein–reagent complex, P* is the labelled-protein, $K_d$ is the dissociation constant and $k_L$ is the rate constant for the labelling process. When [R] > [P], the pseudo-first-order reaction rate ($k_{app}$) is given as follows (see Supplementary Methods)[35]:

$$\frac{[P] + [PR]}{[P]_0} = \exp\left(-k_{app}t\right) \qquad (2)$$

$$k_{app} = \frac{k_L}{1 + K_d/[R]} \qquad (3)$$

According to Eq. (3), a Michaelis-Menten-type saturation curve should be obtained from the $k_{app}$-[R] plot.

The reaction time courses of FKBP12 labelling with various concentrations of **1** (200–2000 nM) were monitored by MALDI-TOF MS (Fig. 3b) and plotted to determine $k_{app}$ using Eq. (2) (Fig. 3c). The values of $k_{app}$ obtained in this manner were then plotted against the concentration of **1**, which were well fitted by Eq. (3) (Fig. 3d). The labelling rate constant $k_L$ and the dissociation constant $K_d$ of **1** were determined from curve fitting analysis to be $(6.1 \pm 0.6) \times 10^{-3}\,s^{-1}$ and $(2.1 \pm 0.2) \times 10^{-7}\,M$, respectively (the mean of triplicate ± standard deviation (s.d)) (Table 1). The obtained $K_d$ value was ~1 order of magnitude higher than the reported $K_d$ of SLF for FKBP12 ($K_d = 20$ nM)[32], which indicates that the derivatization of SLF leads to slightly drop the affinity but does not appreciably reduce the binding ability as reported by other groups[32].

We then performed a kinetic characterisation of other LD chemistries that we have previously developed, including LDT, LDAI and LDBB (Fig. 3a). We evaluated the rate constants of **5** (LDT reagent for FKBP12), **6** (LDAI reagent for eDHFR) and **7** (LDBB reagent for eDHFR) that had been structurally optimised for their target proteins in previous studies[13,18], and compared these kinetic parameters with the LDNASA reagents **1** and **4**. In the case of FKBP12 labelling, the $k_L$ value of LDNASA **1** was 635-fold higher than that of LDT **5** $[(9.6 \pm 0.2) \times 10^{-6}\,s^{-1}]$, while the obtained $K_d$ value of **1** was almost equivalent to **5** (Table 1, Supplementary Figure 7). In an evaluation of labelling reagents for eDHFR, the $k_L$ value of LDNASA **4** was 2241-fold and 38-fold higher than those of LDAI **6** $[(5.8 \pm 0.1) \times 10^{-6}\,s^{-1}]$ and LDBB **7** $[(3.4 \pm 0.3) \times 10^{-4}\,s^{-1}]$, respectively, without significant differences in the affinities of these reagents (Table 1, Supplementary Figure 8). These quantitative kinetic parameters clearly showed that the NASA reactive group had the largest $k_L$ of the reactive groups developed so far.

It is conceivable that the $k_L/K_d$ value corresponds to the second-order rate constant for the labelling reaction in LD

chemistry. Therefore, we can now directly compare the kinetics of LD chemistries with that of other conventional bioorthogonal reactions for protein chemical labelling (Fig. 3e). The second-order rate constants ($k_L/K_d$) for LDT **5**, LDAI **6** and LDBB **7**, which represent the first, second and third-generations of LD chemistry, were estimated to be $45 \pm 6\,M^{-1}\,s^{-1}$, $11 \pm 1\,M^{-1}\,s^{-1}$ and $590 \pm 120\,M^{-1}\,s^{-1}$, respectively. These values are in the same order of magnitude as the rate constants for a widely-used copper catalysed azide-alkyne cycloaddition (CuAAC), tetrazine-cyclopropene cycloaddition, and photoinduced tetrazole-alkene cycloaddition (photo-click ligation)[5,6,10]. On the other hand, LDNASA reagents displayed much higher rate constants, $(2.9 \pm 0.4) \times 10^4\,M^{-1}\,s^{-1}$ for FKBP12 labelling with **1**, and $(9.3 \pm 1.5) \times 10^3\,M^{-1}\,s^{-1}$ for eDHFR labelling with **4**. It is noteworthy that these rate constants ($\sim10^4\,M^{-1}\,s^{-1}$) are almost comparable to the fastest bioorthogonal protein modifications using IEDDA cycloadditions between tetrazines and trans-cyclooctenes, and commonly used enzymatic labelling methods, such as the SNAP/CLIP-tag system, highlighting the rapid kinetics of LDNASA chemistry[8–10].

While the intrinsic kinetics of conventional bioorthogonal reactions primarily relies on their reactivity, that of LD chemistries exploiting the proximity-driven reaction via protein–ligand recognition should strongly depend on the affinity of the reagent for the target proteins, as well as the reactivity of the reactive group. However, it has not quantitatively been defined how much the labelling rate is affected by the ligand affinity, and how strong an affinity is required to achieve efficient and selective protein labelling. To clarify the relationship between the ligand affinity and the reaction profiles in LD chemistry, we subsequently sought to quantify the kinetic parameters of NASA reagent **8** containing LLF, a low-affinity ligand for FKBP12 (reported $K_d = 3.5\,\mu M$)[36]. Prior to the kinetic analysis, we confirmed that the major labelling site of FKBP12 with **8** (Lys 44) is identical to **1** (Supplementary Figure 4). As shown in Supplementary Figure 9, higher concentrations of **8** (μM to sub-mM) were required for efficient labelling in comparison with **1**, which is because of the low binding affinity of LLF. The kinetics assay revealed that the $k_L$ value of **8** $[(7.6 \pm 1.7) \times 10^{-3}\,s^{-1}]$ is almost the same as that of **1** (Table 1). In stark contrast, the second-order rate constant $k_L/K_d$ of **8** ($74 \pm 19\,M^{-1}\,s^{-1}$) was 392-fold less than **1**, because of its large $K_d$ value $[(1.0 \pm 0.4) \times 10^{-4}\,M]$. These results clearly demonstrated that the affinities of the ligand of LD reagents predominantly determines the labelling kinetics when the reactive group is identical, and also indicated that a ligand with sub μM order affinity is required to achieve rapid and efficient labelling with LDNASA chemistry.

**Selective protein labelling in crude biological contexts.** In addition to the reaction rate and efficiency, target selectivity is also important for the biological application of protein chemical labelling. To verify the selectivity and bioorthogonality of LDNASA chemistry, we next conducted protein labelling

**Table 1 Kinetic and binding parameters of labelling reagents for FKBP12 and eDHFR**

| Protein | Reagent | $k_L$ (s$^{-1}$) | $K_d$ (M) | $k_L/K_d$ (M$^{-1}$s$^{-1}$) |
|---|---|---|---|---|
| FKBP12 | **1** (LDNASA) | $(6.1 \pm 0.6) \times 10^{-3}$ | $(2.1 \pm 0.2) \times 10^{-7}$ | $(2.9 \pm 0.4) \times 10^4$ |
| | **5** (LDT) | $(9.6 \pm 0.2) \times 10^{-6}$ | $(2.1 \pm 0.3) \times 10^{-7}$ | $45 \pm 6$ |
| | **8** (LDNASA) | $(7.6 \pm 1.7) \times 10^{-3}$ | $(1.0 \pm 0.4) \times 10^{-4}$ | $74 \pm 19$ |
| eDHFR | **4** (LDNASA) | $(1.3 \pm 0.1) \times 10^{-2}$ | $(1.4 \pm 0.2) \times 10^{-6}$ | $(9.3 \pm 1.5) \times 10^3$ |
| | **6** (LDAI) | $(5.8 \pm 0.1) \times 10^{-6}$ | $(5.5 \pm 0.3) \times 10^{-7}$ | $11 \pm 1$ |
| | **7** (LDBB) | $(3.4 \pm 0.3) \times 10^{-4}$ | $(5.8 \pm 1.1) \times 10^{-7}$ | $(5.9 \pm 1.2) \times 10^2$ |

The data represent the mean of triplicate ± standard deviation

experiments in different biological environments. First, FKBP12 labelling in cell lysate was carried out using **1** and **8** to investigate whether the affinity of LD reagents affects the labelling efficiency and target selectivity in crude conditions. HeLa cell lysate containing 1 μM of FKBP12 was incubated with **1** (1 μM) and **8** (1–20 μM) for 1 h at 37 °C, and analysed by SDS-PAGE and western blotting using streptavidin–horseradish peroxidase conjugate (SAv–HRP). As shown in lane 2 of Fig. 4a, a specific band (12 kDa) corresponding to the biotinylated FKBP12 was clearly observed using **1**. Such selective FKBP12 labelling was inhibited in the presence of rapamycin (lane 3), and instead, some labelling reactions with proteins other than FKBP12 occurred owing to the presence of **1** in the free (unbound) state. We also conducted a titration experiment with higher concentrations of **1** (1–20 μM) to the cell lysate (Supplementary Figure 10). Although several bands due to unspecific labelling appeared in the ratio of reagent-to-protein greater than one, the labelling band of FKBP12 was predominant (the ratio of unspecific to specific labelling is less than 0.15). When the LDNASA reagent bearing the weak affinity ligand LLF (**8**) was used, the FKBP12 labelling was substantially diminished (lane 4 of Fig. 4a). To obtain the same signal intensity as biotinylated FKBP12 labelled by **1**, 20 μM of **8** was required (lane 6 of Fig. 4a). However, such a high concentration of the

LDNASA reagent caused non-specific labelling reactions with many non-targeted proteins. Given the affinity of **1** [$K_d = (2.1 \pm 0.2) \times 10^{-7}$ M], these results clearly indicate that the ligand affinity of the LDNASA reagent requires at least a sub μM order $K_d$ value to ensure sufficient bioorthogonality (target selectivity) in crude environments.

We then conducted intracellular endogenous FKBP12 labelling. Mouse myoblast C2C12 cells were incubated in culture medium containing **1** for 10–120 min at 37 °C. The cells were lysed and analysed by western blotting. As shown in lanes 2–5 of Fig. 4b, endogenous FKBP12 inside live cells was specifically modified with biotin using LDNASA **1** and this labelling was abolished on co-incubation with rapamycin (lane 6 of Fig. 4b), indicating that recognition-driven protein labelling efficiently proceeded inside the live cells. The time course of intracellular FKBP12 labelling with LDNASA **1** is shown in Fig. 4c. Although the reaction rate was slightly slower than that of the in vitro experiments, probably because of the low membrane-permeability of **1**, it should be noted that the labelled band was detectable after just 10 min incubation with **1**, highlighting the rapid kinetics of LDNASA chemistry. Quantitative western blotting analysis revealed the labelling yield with LDNASA **1** at 60 min to be 78% of the entire population of FKBP12 (almost quantitative at 120 min) (Fig. 4c

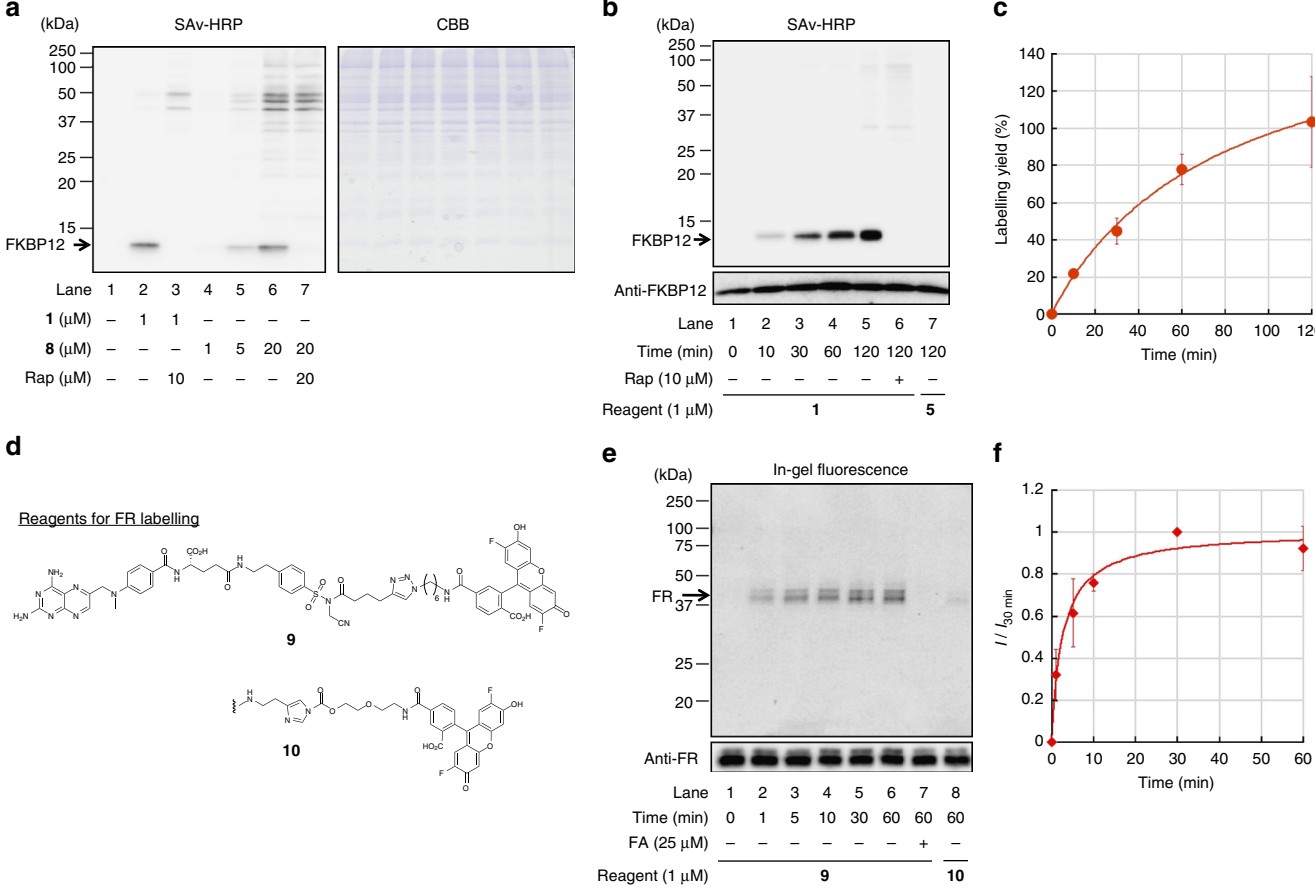

**Fig. 4** Target-selective protein labelling with LDNASA chemistry. **a** Western blotting analysis of FKBP12 labelling in cell lysates. HeLa cell lysate was mixed with recombinant FKBP12 (1 μM) and incubated with reagent in 50 mM HEPES buffer, pH 7.2, 37 °C, 1 h. SAv-HRP, streptavidin–horseradish peroxidase conjugate; CBB, Coomassie brilliant blue staining. The high background (smear) signals of lane 6 and 7 may suggest unspecific labelling due to a high concentration of reagent (20 μM), whereas such background signals were less in lanes 1–5 (1–5 μM of reagent). **b** Western blotting analysis of endogenous FKBP12 labelling in live C2C12 cells. The cells were treated with **1** or **5** (1 μM) for 0–120 min at 37 °C in medium. **c** Time course plot of the endogenous FKBP12 labelling with **1**. **d** Molecular structures of LDNASA **9** and LDAI **10** for FR labelling. **e** In-gel fluorescence and western blotting analysis of endogenous FR labelling in KB cells. The cells were treated with **9** or **10** (1 μM) for 0–60 min at 37 °C in medium. FA, Folic acid (competitive inhibitor). **f** Time profile of the endogenous FR labelling with **9**. $I$ and $I_{30\,min}$ are band intensities of Oregon Green-labelled FR at the indicated time point and at 30 min, respectively. Error bars represent s.d., $n = 3$

and Supplementary Figure 11), while the LDT reagent 5, which has successfully achieved intracellular FKBP12 labelling in our previous work[13], failed to label FKBP12 in C2C12 cells in that time frame (<120 min) (lane 7 of Fig. 4b).

The applicability of LDNASA chemistry to membrane proteins was demonstrated by folate receptor (FR) labelling in KB cells with LDNASA 9 containing the methotrexate (MTX) ligand ($K_d \approx 200$ nM for FR)[37] and an Oregon-Green probe (Fig. 4d). The specific fluorescent band of the labelled-FR was observed in in-gel fluorescence detection (Fig. 4e), and the labelling reaction was completed within 30 min using LDNASA 9 (Fig. 4f), while negligible signal was observed with the previous LDAI reagent 10 in such a short time. Overall, these results demonstrated that LDNASA reagents can rapidly label both endogenous intracellular and membrane-associated proteins in a highly specific manner in native multi-molecular crowding biological contexts.

**Design of a NASA-based irreversible inhibitor for Hsp90.** LD reagents are normally designed to incorporate a synthetic probe into a protein, but conversely, they can also be used to attach the ligand moiety (instead of the probe) to a protein by switching the linkage direction. We expected that such a LD reagent may act as an irreversible inhibitor of the target protein[22,23]. As demonstrated above, LDNASA chemistry allows rapid and efficient covalent modification of canonical nucleophilic amino acids (e.g.

the ε-amino group of non-catalytic lysine) of the target (endogenous) protein. Given these unique properties, which cannot be addressed by other bioorthogonal methods, we envisioned that NASA-based protein labelling could serve as a useful strategy to develop a selective covalent inhibitor, in which the NASA group is used as a warhead for the covalent bond formation between a ligand and the target protein. As a proof-of-principle, we chose Hsp90 as a target protein[38,39]. Recent studies have revealed that the chaperone activity of Hsp90 is closely associated with the pathology of various diseases, including tumourigenesis, inflammation and neurodegenerative diseases. Therefore, Hsp90 is now considered an attractive drug target, and covalent inhibition of its chaperone activity holds promise as a therapeutic strategy for treatment of these conditions[40].

Prior to the development of a covalent inhibitor for Hsp90, we investigated whether the NASA-based reagents can selectively react with endogenous Hsp90 in live cells. We designed the LDNASA reagent 11, which consists of PU-H71[41,42], a specific (reversible) ligand for the N-terminal ATP binding domain of Hsp90, and fluorescein diacetate (AcFL) as the detection probe (Fig. 5a). It was revealed that LDNASA 11 labelled endogenous Hsp90 (both α and β isoforms) in breast cancer SKBR3 cells after 3 h with sufficient selectivity even using high concentrations of the reagent (~10 μM) (Fig. 5b, Supplementary Figures 12 and 13). This reaction was inhibited by various Hsp90 inhibitors that

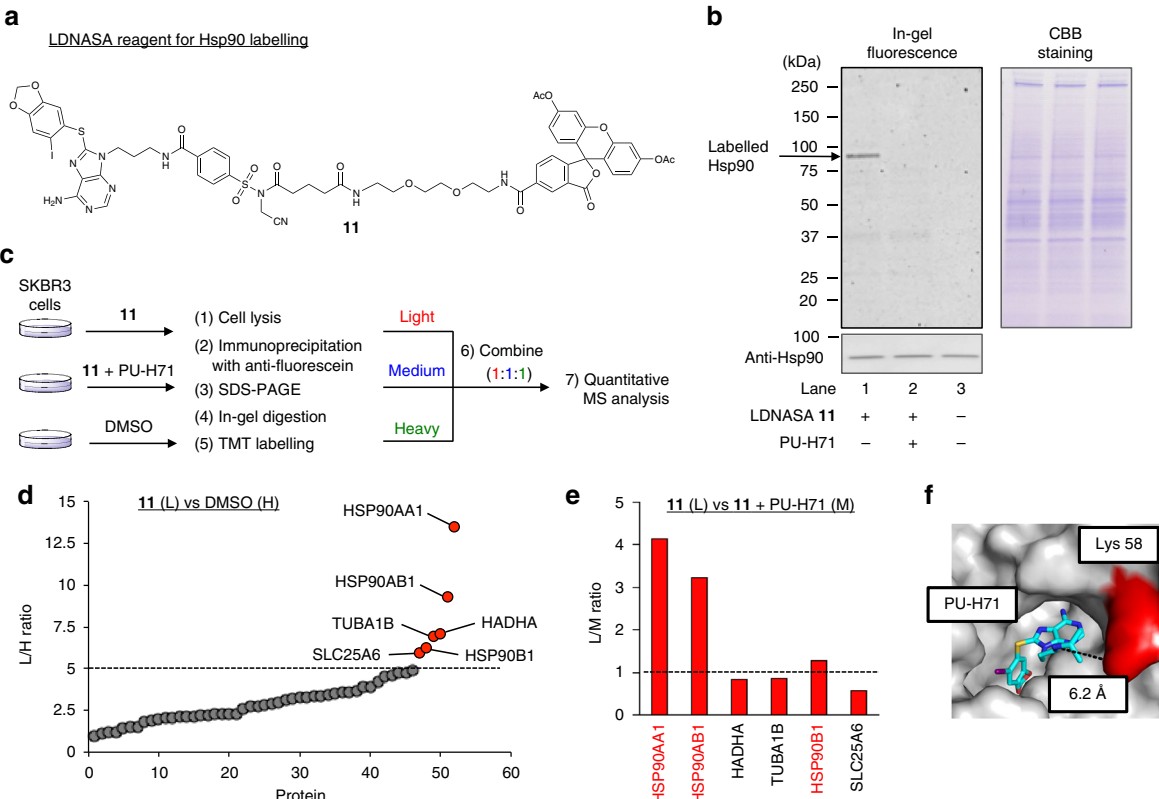

**Fig. 5** Selective and site-specific labelling of endogenous Hsp90 in live cells. **a** Molecular structure of LDNASA 11 for Hsp90 labelling. **b** SDS-PAGE and western blotting analysis of the labelling reaction in live SKBR3 cells. The cells were treated with 11 (0.5 μM) in the absence or presence of PU-H71 (10 μM) for 3 h at 37 °C in medium (pH 7.4). After washing, the cells were lysed and analysed by in-gel fluorescence and western blotting using anti-Hsp90 antibody. **c** Workflow for TMT-based quantitative LC-MSMS analysis of proteins labelled with 11. After in-gel digestion, obtained peptide fragments were modified with Light- (L-), Medium- (M-), or Heavy- (H-) TMT reagent for samples treated with 11 (0.5 μM, 3 h), 11 with PU-H71 (10 μM), or DMSO, respectively. **d** L/H ratio plots for total proteins identified in experiments comparing cells treated with 11 versus DMSO. Proteins with median L/H ratios >5 are assigned as 11-labelled proteins (red plots). **e** L/M ratios (11 versus 11 with PU-H71) of proteins assigned as 11-targets. Gene names of ligand-specific targets (L/M ratios >1) and off-targets (L/M ratios <1) are written in red and black, respectively. **f** The crystal structure of the N-terminal ATP binding domain of Hsp90α–PU-H71 complex (PDB ID: 2FWZ). The residue (Lys 58) modified with 11 and 12 is highlighted in red, and the PU-H71 ligand is coloured in blue

competitively bind the ATP-binding site, indicating the labelling occurred by an affinity-driven proximity effect (Supplementary Figure 14).

While these gel based analyses showed the selective labelling of Hsp90, we more carefully evaluated off-target proteins of 11 in live cells by a quantitative mass spectrometry using tandem mass tag (TMT) labelling (Fig. 5c)[43]. The labelled proteins were enriched by immunoprecipitation with an anti-fluorescein antibody and tryptically digested in gel. Digested peptides were modified with Light- (L-), Medium- (M-), or Heavy- (H-) TMT reagent for samples treated with 11, 11 with PU-H71 (as competitive condition), or DMSO (vehicle), respectively, and mixed in a 1:1:1 ratio for LC-MSMS analysis. We performed three experimental replicates, in which proteins detected and quantified at least twice were selected as identified proteins. According to the related approach of Cravatt et al[44], the proteins having L/H (11/DMSO) ratios >5 were defined as 11-labelled proteins. We also designated the ligand-specific targets or off-targets as proteins exhibiting L/M (11/11 with PU-H71) ratios more or less than 1, respectively. On the basis of these criteria, we identified 6 proteins that react with 11, including three ligand-specific targets (Hsp90α (HSP90AA1), Hsp90β (HSP90AB1), and Grp94 (HSP90B1)) and three off-targets (Trifunctional enzyme subunit α (HADHA), Tubulin-α (TUBA1B), and ADP/ATP translocase 3 (SLC25A6)) (Fig. 5d, e and Supplementary Data 1). Notably, the targeted Hsp90α and Hsp90β showed the highest L/H ratios among identified proteins (13.5 and 9.3, respectively). Grp94, a HSP90 isomer, was previously reported to be a PU-H71 specific target. TRAP-1, another Hsp90 isomer that can bind PU-H71, on the other hand, was not identified as 11-target by our criteria[45]. Both the gel-based analysis and chemoproteomic data clearly demonstrated the high selectivity of the NASA reagent for Hsp90, although there are a few off-target proteins.

The labelling site of cellular Hsp90α was identified to be Lys58 located at the entrance of the ligand-binding pocket of the ATP binding domain of Hsp90α (Fig. 5f and Supplementary Figure 15). On the basis of the crystal structure (PDB ID: 2FWZ), the labelled lysine residue of Hsp90α, which is conserved in Hsp90β, is close to the isopropyl amine moiety of the PU-H71 ligand at a distance of approximately 6.2 Å (Fig. 5f and Supplementary Figure 16).

Given all of these data, we rationally designed the NASA-based covalent inhibitor 12, in which a PU-H71 ligand is connected to a NASA warhead with a 6.3 Å–linker (Fig. 6a, b). The quantitative and single covalent attachment of the PU-H71 ligand was confirmed to be Lys 58, identical to the labelling site with LDNASA 11, using a recombinant N-terminal ATP binding domain of Hsp90 and 12 in test tube experiments (Fig. 5f, Supplementary Figures 17 and 18). The kinetic study gave us the affinity ($K_i$) of 12 for Hsp90 to be 62 nM, which is in good agreement with the reported $IC_{50}$ value (~50 nM) of PU-H71 ligand (Supplementary Figure 19)[41]. Also, the second-order rate constant of the covalent inhibition was estimated to be $2.9 \times 10^4$ $M^{-1} s^{-1}$ equivalent to that of LDNASA 1 (Table 1). The covalent bond formation of 12 with intracellular Hsp90 in live cells was next evaluated by an inhibition experiment in which the fluorescein labelling ratio of Hsp90 with 11 in the presence of 12 or PU-H71 was compared. SKBR3 cells were pre-treated with PU-H71 or 12 (1–10 μM) for 3 h, followed by extensive washing with culture medium to remove unbound inhibitors, and then labelled with 11 (0.5 μM). As shown in Fig. 6c, pre-treatment with compound 12 strongly inhibited the fluorescent labelling. In contrast, non-covalent PU-H71 failed to block the labelling reaction under such washing conditions. These results clearly indicate the durable occupancy of compound 12 in the binding site of intracellular Hsp90 because of the covalent bond, whereas the noncovalent (reversible) inhibitor PU-H71 was readily

washed out resulting in a loss of potency. These results encouraged us to examine the potential proliferation-inhibiting effects of compound 12 on tumour cells. SKBR3 cells were treated with PU-H71 or 12 for 3 h, followed by washing with medium. The cells were further incubated in medium for 69 h, and the cell viability was evaluated with the WST-8 assay[46]. While the long-term incubation of the cells with PU-H71 (for 72 h) showed remarkable cytotoxicity, treatment with PU-H71 for only 3 h almost abolished the cytotoxicity due to the rapid washout from cells (Fig. 6d and Supplementary Figure 20). On the other hand, 3 h incubation of compound 12 clearly showed a growth inhibitory effect at a concentration of 1–10 μM (Fig. 6d). We also examined the expression levels of client proteins of Hsp90 (Fig. 6e, f). As previously reported[41], when PU-H71 (1 μM) was treated for 24 h (without washing), the western blotting bands of major Hsp90-client proteins, such as HER2, cRaf, Akt and pAkt, dramatically decreased relative to the vehicle control, which was reasonably ascribed to the inhibition of the chaperone activity of Hsp90 by PU-H71 (Supplementary Figure 21). Such a decrease in Hsp90-client proteins was also observed in the cells that were treated with 12 for 3 h followed by washing and further incubating for 21 h (lanes 3–5 of Fig. 6e), whereas PU-H71 did not significantly affect the band intensities of client proteins under the same washing conditions (lanes 1 and 2). These results explicitly demonstrated that the NASA-based covalent inhibitor 12 is a potent long-term (irreversible) inhibitor of the chaperone activity of intracellular Hsp90, which is in agreement with the observed cytotoxicity effect of 12.

## Discussion

In summary, we have developed LDNASA chemistry that shows rapid reaction kinetics with sufficient target selectivity and bioorthogonality under native live cell conditions. The second-order rate constant was comparable to that of the fastest bioorthogonal protein modification methods. We also found that the NASA reactive group is capable of labelling the ε-amino group of non-catalytic Lys residues. It is generally considered that ~99.9% of the Lys residues on protein surface are protonated, which suppresses their nucleophilicities (reactivities) under physiological pH[27]. However, our data strongly suggest that the chemical labelling of the Lys residues can proceed by the close proximity of the NASA reactive group to the ε-amino group of Lys. While the ligand-directed chemistry requires an appropriate ligand and may suffer to some extent from the unspecific labelling in biological crude environments, the present study indicated that this limitation can be addressed by careful design of reagents (use of a ligand with a $K_d$ value of sub μM order) and thorough optimisation of reaction conditions. Moreover, we demonstrated that the mass spectroscopy-based analysis is quite powerful for the proteome-wide characterisation of the possible off-target proteins in this method. These results provide deep insights and important guidelines for practical use of not only LDNASA chemistry but also other proximity-driven protein labelling in living systems.

We also demonstrated that NASA-based protein modification is applicable for the irreversible suppression of native protein functions in living cells. To the best of our knowledge, this study is the first report of covalent inhibition of the N-terminal ATP binding domain essential for the catalytic chaperone activity of Hsp90. Inhibition of the ATPase activity in an irreversible fashion should be particularly effective in enhancing the drug potency inside cells, where a high concentration of endogenous ATP/ADP (1–10 mM) may competitively attenuate the potency of reversible inhibitors[47]. Furthermore, the present study clearly showed that the NASA warhead is able to react with a lysine residue that is close to the ligand binding site, but not the catalytic residue.

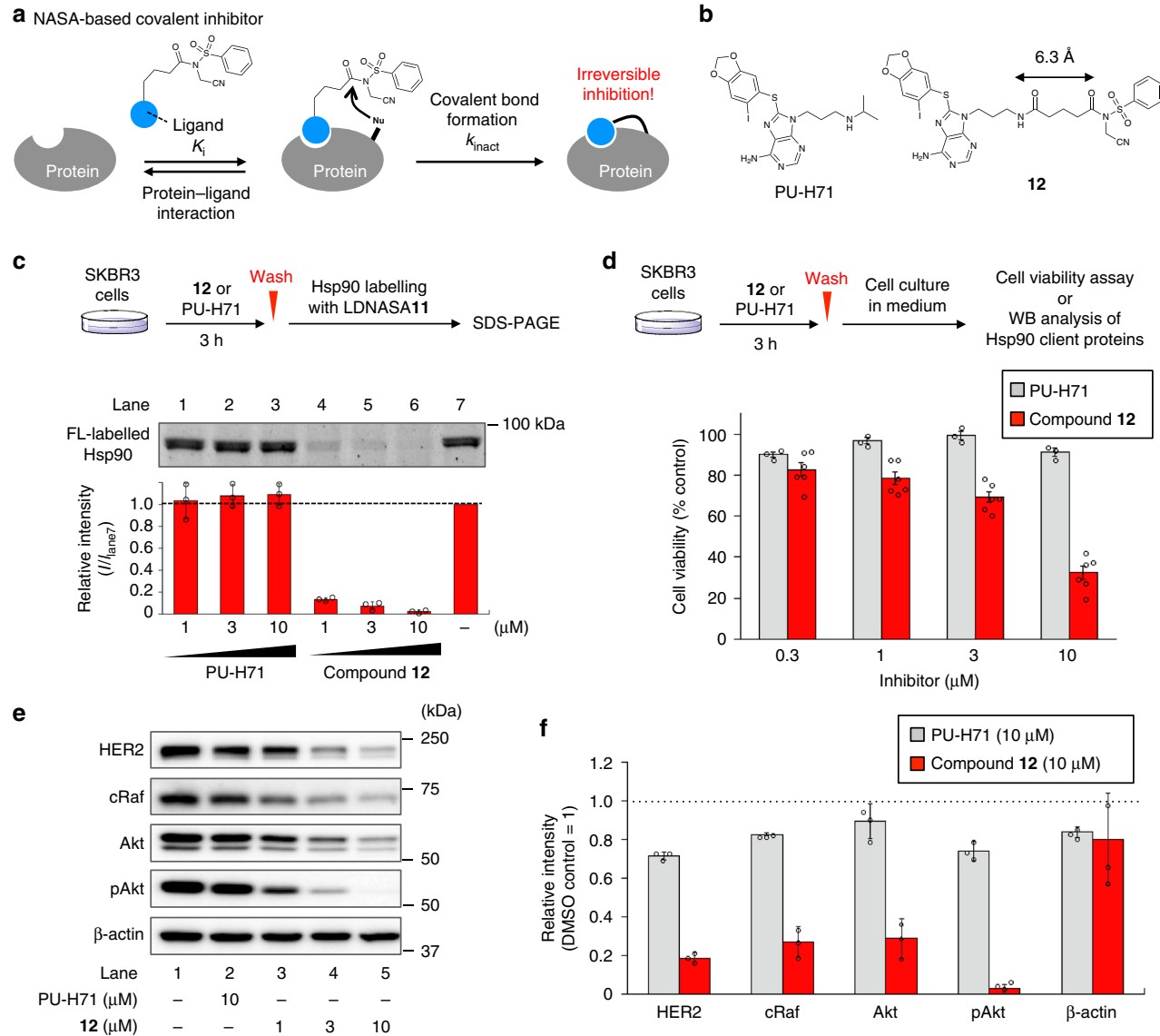

**Fig. 6** Irreversible inhibition of intracellular Hsp90. **a** Schematic illustration of reaction mechanism of NASA-based covalent inhibitor. **b** Molecular structures of PU-H71 (non-covalent inhibitor) and NASA-based covalent inhibitor **12**. **c** In-gel fluorescence analysis of fluorescein labelling of Hsp90 with **11** in SKBR3 cells after inhibitor washout. After cells were incubated with various concentrations of PU-H71 or **12** for 3 h, the cells were washed with medium and further incubated with **11** for 3 h. The cells were washed, lysed and analysed by in-gel fluorescence imaging. Error bars represent s.d., n = 3. **d** Viability of the cells 69 h after PU-H71 or **12** washout. Data represent mean values ± standard error of the mean (s.e.m.) for three (PU-H71) or six (compound **12**) independent experiments. **e** Western blotting analysis of the destabilization of client proteins induced by inhibiting Hsp90 chaperone activity, and **f** the normalised band intensities. Cells were treated with PU-H71 or **12** for 3 h, followed by washing with media and further incubation for 21 h. The protein band intensity was normalised to DMSO control (lane 1). β-actin is a control as a non-Hsp90 client protein. Error bars represent s.d., n = 3

There have been few surface lysine-targetable inhibitors developed to date[27,48–50], and in general, it is quite difficult to rationally design such an inhibitor, mainly because of the lack of suitable warheads for such less-reactive residues[51]. In this respect, the NASA reactive group should have great potential as a versatile electrophile for lysine-targeted irreversible inhibition and is expected to expand the scope of targetable proteins in covalent drug development.

## Methods

**Synthesis**. All synthetic procedures and compound characterisations are described in Supplementary Methods.

**General materials for biological experiments**. All biological reagents were purchased from Sigma-Aldrich, Tokyo Chemical Industry (TCI), Wako Pure Chemical Industries, Sasaki Chemical, Bio-Rad, Thermo Fisher Scientific, Nacalai Tesque or

Watanabe Chemical Industries, and used without further purification, unless otherwise noted. Sodium dodecyl sulfate-polyacrylamide gel electrophoresis (SDS-PAGE) and western blotting were carried out with a Bio-Rad Mini-Protean III electrophoresis apparatus. Fluorescence gel images and chemical luminescent signals using Chemi-Lumi one (Nacalai Tesque) or ECL Prime (GE Healthcare) were obtained by an imagequant LAS4000 (Fujifilm). Matrix-assisted laser desorption/ionisation time-of-flight mass spectrometry (MALDI-TOF MS) spectra were measured with an Autoflex III, Ultraflex III or UltrafleXtreme (Bruker Daltonics) using α-cyano-4-hydroxycinnamic acid (CHCA) or sinapic acid as the matrix. UV-vis absorption spectra were obtained with a Shimadzu UV-2550 spectrophotometer. Reversed-phase HPLC (RP-HPLC) was carried out on a Hitachi Chromaster system equipped with a 5430 UV-vis detector, a 5440 fluorescence detector, and a YMC-Pack ODS-A column (5 µm, 250 × 4.6 mm) at a flow rate of 1.0 mL/min. All runs used linear gradients of acetonitrile containing 0.1% trifluoroacetic acid (TFA) and 0.1% aqueous TFA.

**FKBP12 labelling in vitro**. Recombinant human FKBP12 was obtained as described in Supplementary Methods. Purified FKBP12 (5 µM) was incubated with

reagent (10 μM) in the absence or presence of rapamycin (20 μM) in HEPES buffer (50 mM, pH 7.2) at 37 °C. Aliquots at different time points were taken and then desalted using a Ziptip-C4 (Merck), and the labelling yields were determined by MALDI-TOF MS (matrix: CHCA).

**Peptide mapping of the Bt-labelled FKBP12.** Purified FKBP12 (39.4 μM) was incubated with LDNASA **1** or **8** (39.4 μM) in HEPES buffer (50 mM, pH 7.2) at 37 ° C. After 0.5 h, the labelled FKBP12 was purified by size-exclusion chromatography using a TOYOPEARL HW-40F column with pH 8.0 50 mM HEPES buffer. To this solution, urea (at a final concentration of 2 M) and Trypsin (Trypsin/substrate ratio = 1/5 (w/w)) were added. After incubation at 37 °C for 24 h, the digested peptides were separated by analytical RP-HPLC. The collected fractions were analysed by MALDI-TOF MS (matrix: CHCA) and the labelled fragment was further characterised by MALDI-TOF-TOF MS/MS analysis. To determine the minor labelling site, the digested peptides were desalted using a Ziptip-C18 (Merck) and analysed by nanoflow reverse liquid chromatography followed by tandem MS, using a LTQ Orbitrap XL hybrid mass spectrometer (Thermo Fisher Scientific). For LTQ-Orbitrap XL, a capillary reverse phase HPLC-MS/MS system composed of an Agilent 1100 series gradient pump equipped with Calco C2 valves with 150-μm ports, and LTQ-Orbitrap XL hybrid mass spectrometer equipped with an XYZ nanoelectrospray ionisation (NSI) source (AMR). Samples were automatically injected using PAL system (CTC analytics, Zwingen, Switzerland) into a peptide L-trap column OSD (5 μm, AMR) attached to an injector valve for desalinating and concentrating peptides. After washing the trap with MS-grade water containing 0.1% trifluoroacetic acid and 2% acetonitrile (solvent C), the peptides were loaded into a nano HPLC capillary column (C18 packed with the gel particle size of 3 μm, 0.1 × 150 mm, Nikkyo Technos, Tokyo Japan) by switching the valve. Water containing 0.1% formic acid (solvent A) and 80% acetonitrile containing 0.1% formic acid (solvent B) were used as eluents. The column was developed with the concentration gradient of acetonitrile as follows: from 5% B to 35% B in 150 min, from 45% B to 95% B in 1 min, sustaining 95% B for 20 min, from 95% B to 5% B in 1 min, and finally re-equilibrating with 5% B for 8 min. The Xcalibur 2.1 system (Thermo Fisher Scientific) was used to record peptide spectra over the mass range of $m/z$ 350–1500, and MS/MS spectra in information-dependent data acquisition over the mass range of $m/z$ 150–2000. Repeatedly, MS spectra were recorded followed by two data-dependent collision induced dissociation (CID) MS/MS spectra generated from two highest intensity precursor ions. Multiple charged peptides were chosen for MS/MS experiments due to their good fragmentation characteristics. MS/MS spectra were interpreted, and peak lists were generated by Proteome Discoverer 1.4.1.14 (Thermo Fisher Scientific). Searches were performed using the SEQUEST HT (Thermo Fisher Scientific) against the latest uniprot database for whole Human proteome (Homo sapiens, 9606). The database was searched for tryptic-digested peptides with up to three miscleavages. Dynamic modifications of biotin on any amino acids and oxidation on methionine were searched with peptide mass tolerance at 10 ppm and MS/MS mass tolerance of 0.6 Da. The resultant data set was filtered to a maximum false discovery rate (FDR) of 0.01.

**eDHFR labelling in vitro.** Recombinant eDHFR was obtained as described in Supplementary Methods. Purified eDHFR (10 μM) was incubated with **4** (20 μM) in the absence or presence of trimethoprim (TMP) (50 μM) in HEPES buffer (50 mM, pH 7.2) at 37 °C. Aliquots at different time points were taken and then desalted using a Ziptip-C4, and the labelling yields were determined by MALDI-TOF MS (matrix: sinapic acid)

**Peptide mapping of the Dc-labelled eDHFR.** Recombinant eDHFR (50 μM) was incubated with LDNASA **4** (50 μM) in HEPES buffer (50 mM, pH 7.2) at 37 °C for 1 h. The Dc-labelled eDHFR was purified by size-exclusion chromatography (TOYOPEARL HW-40F column, TOSOH) with pH 8.0 50 mM HEPES buffer. The protein was denatured with urea (at a final concentration of 4 M), and then treated with Trypsin (Trypsin/substrate ratio = 1/10 (w/w)) at 37 °C for 58 h. The digested peptides were separated by RP-HPLC with UV (absorbance at 220 nm) and fluorescent detector ($\lambda_{ex}$ = 430 nm, $\lambda_{em}$ = 480 nm), followed by the analysis of the collected fractions by MALDI-TOF MS and MS/MS analysis.

**Evaluation of stabilities of LDNASA reagents.** LDNASA **1–3** (10 μM) and internal standard (Trimethoprim, 10 μM) were incubated in HEPES buffer (50 mM, pH 7.2) at 37 °C. Samples at different time points were taken and analysed by RP-HPLC.

**Kinetic analysis of FKBP12 labelling.** Purified FKBP12 (100 nM) was incubated with reagent (200–2000 nM) in HEPES buffer (50 mM, pH 7.2) at 37 °C. Aliquots at different time points were taken, and the reaction was immediately quenched using a Ziptip-C4. The residual ratio of nonlabelled protein ([P] + [PR]/[P]$_0$) were determined by MALDI-TOF MS (matrix: CHCA). The pseudo-first order reaction rates ($k_{app}$) were obtained by fitting the data to Eq. (2). Kinetic parameters were obtained by fitting the data to Eq. (3).

**Kinetic analysis of eDHFR labelling.** Purified eDHFR (300 nM) was incubated with reagent (0.6–6 μM) in HEPES buffer (1 mM, pH 7.2) at 37 °C. Aliquots at different time points were taken, and the reaction was immediately quenched by mixing acidic matrix solution (10 mg/mL sinapic acid, 0.1% TFA, 50% acetonitrile in water). The residual ratio of nonlabelled protein ([P] + [PR]/[P]$_0$) were determined by MALDI-TOF MS, and the kinetic parameters were obtained by the method described above.

**FKBP12 labelling in HeLa cell lysate.** HeLa cells (kindly gifted from Prof. Yoshiki Katayama) ($1 \times 10^7$ cells) were suspended in HEPES buffer (50 mM, pH 7.2) containing 1% protease inhibitor cocktail set III (Calbiochem) and lysed by Potter-Elvehjem homogeniser at 4 °C. The lysate was centrifuged at 16 000 × g for 3 min, and the supernatant was collected. Protein concentration was determined by BCA assay and adjusted to 0.5 mg/mL. This solution was incubated with recombinant FKBP12 (final concentration 1 μM) and **1** (1 μM) or **8** (1–20 μM) in the absence or presence of Rapamycin (10 or 20 μM) for 1 h at 37 °C. The reaction mixture was mixed with 1/4 volume of 5 × sample buffer (pH 6.8, 312.5 mM Tris–HCl, 25% sucrose, 10% SDS, 0.025% bromophenol blue) containing 250 mM DTT and incubated for 1 h at 25 °C. The samples were analysed by western blotting using Streptavidin-HRP conjugate (SAv-HRP, Thermo, S911, 1:5000) and Coomassie Brilliant Blue (CBB) stain.

**Chemical labelling of endogenous FKBP12 in C2C12 cells.** Mouse myoblast C2C12 cells (ATCC) ($2.0 \times 10^5$ cells) were cultured in Dulbecco's modified Eagle's medium (DMEM) supplemented with 10% foetal bovine serum (FBS, Gibco), penicillin (100 units/ml), streptomycin (100 mg/ml), and amphotericin B (250 ng/ml), and incubated in a 5% CO$_2$ humidified chamber at 37 °C. The cells were then incubated in FBS-free DMEM containing reagent (1 μM) at 37 °C for 120 min. As control experiments, the labelling was conducted in the presence of Rapamycin (10 μM). For western blot analysis, after washing twice with PBS, the cells were lysed with RIPA buffer (pH 7.4, 25 mM Tris–HCl, 150 mM NaCl, 0.1% SDS, 1% Nonidet P-40, 0.25% deoxycholic acid) containing 1% protease inhibitor cocktail set III (Calbiochem). The lysed samples were collected and centrifuged (15 200 × g, 10 min at 4 °C). The supernatants were mixed with 1/4 volume of 5× sample buffer (pH 6.8, 312.5 mM Tris–HCl, 25% sucrose, 10% SDS, 0.025% bromophenol blue) containing 250 mM DTT and incubated for 1 h at 25 °C. The samples were subjected to SDS-PAGE and electro-transferred onto an Immun-Blot PVDF membrane (Bio-Rad). The labelled FKBP12 was detected by chemiluminescence analysis using Streptavidin-HRP conjugate, rabbit anti-FKBP12 antibody (Abcam, ab2918, 1:1000) and anti-rabbit IgG-HRP conjugate (CST, #7074 S, 1:5000).

**Chemical labelling of endogenous FR in KB cells.** KB cells were obtained from Cell research centre for biomedical research (Institute of development, aging and cancer, Tohoku university), tested negative for mycoplasma contamination, and used without further authentication, to demonstrate endogenous FR labelling in live cell contexts because FR is highly expressed in the cells. KB cells ($2.0 \times 10^5$ cells) were cultured in folate-free RPMI1640 (Gibco) supplemented with 10% FBS, penicillin (100 units/ml), streptomycin (100 mg/ml), and amphotericin B (250 ng/ml), and incubated in a 5% CO$_2$ humidified chamber at 37 °C. The cells were washed three times with folate- and FBS-deficient RPMI1640, and then incubated in the medium containing reagent (1 μM) at 37 °C. The labelling was also conducted in the presence of folate (25 μM) as a negative control. After washing three times with PBS, the cells were lysed on ice by RIPA buffer containing 1% protease inhibitor cocktail set III. The lysate was collected into a tube and centrifuged for 10 min at 15 200 × g. The supernatant was mixed with the same volume of 5 × sample buffer and incubated for 1 h at 25 °C. After SDS-PAGE, fluorescence signals in the gel were detected by LAS4000. The expression of FR was confirmed by western blotting analysis using anti-FR antibody (Abcam, ab125030, 1:1000) and anti-rabbit IgG-HRP conjugate.

**Chemical labelling of endogenous Hsp90 in SKBR3 cells.** SKBR3 human breast cancer cells (ATCC) ($4.0 \times 10^5$ cells) were cultured in McCoy's 5 A supplemented with 10% FBS, penicillin (100 units/ml), streptomycin (100 mg/ml), and amphotericin B (250 ng/ml), and incubated in a 5% CO$_2$ humidified chamber at 37 °C. The cells were then incubated in HEPES-modified McCoy's 5 A (FBS-free) containing reagent (indicated concentration) at 37 °C for indicated time. As control experiments, the labelling was conducted in the presence of competitive inhibitors. After labelling, the cells were washed twice with PBS, and lysed with RIPA buffer. The lysed sample was collected and centrifuged (15 200 × g, 10 min at 4 °C). The supernatant was mixed with 1/4 volume of 5 × sample buffer and vortexed for 1 h at room temperature. The samples were subjected to in-gel fluorescence and western blotting analysis using anti-Hsp90 antibody HRP conjugate (CST, #79641, 1:1000).

**Identification of 11-labelled proteins in live cells.** SKBR3 human breast cancer cells ($2.0 \times 10^6$ cells) were seeded on a 10 cm dish and cultured for 3 days in McCoy's 5A in a 5% CO$_2$ humidified chamber at 37 °C. The cells were then incubated in HEPES-modified McCoy's 5A (FBS-free) containing **11** (0.5 μM) or DMSO (vehicle) at 37 °C for 3 h. As competitive experiments, the labelling was

conducted in the presence of PU-H71 (10 μM). After labelling, the cells were washed twice with PBS, and lysed with RIPA buffer. The lysed sample was collected and centrifuged (15 200 × $g$, 10 min at 4 °C). The supernatant was mixed with chilled acetone and incubated overnight at −80 °C. The protein precipitates were solubilized by sonication in 1 mL of RIPA buffer containing 1% SDS, and then 10-fold diluted with RIPA buffer to reduce SDS concentration to c.a. 0.1%. The protein solution (containing 1 mg of protein) was mixed with Protein G Sepharose 4 Fast Flow (GE Healthcare) and rotated at 4 °C for 1 h. After removal of the beads, the remaining supernatant was mixed with anti-fluorescein antibody (Abcam, ab19491, 7:10000) and rotated at 4 °C for 1 h, followed by addition of fresh Protein G Sepharose 4 Fast Flow and further incubation at 4 °C for 4 h. The beads were washed five times with RIPA buffer and once with PBS. Proteins were eluted from the beads by addition of 2 × sample buffer containing 100 mM DTT and boiling at 95 °C for 5 min. The samples were subjected to a 7.5% SDS–PAGE gel (Mini-PROTEAN TGX Gels, BioRad), and electrophoresed as the migration distance of dye front is about 10 mm. Each lane was excised (height, 10 mm), and treated with a 47.5% methanol / 47.5% pure water / 5% acetic acid for 20 min, a 50% methanol / 50% pure water for 10 min, and in pure water for 10 min. The gel pieces were dehydrated with 100% acetonitrile, and then swelled with 200 μL of 100 mM TEAB (triethylamine bicarbonate) buffer containing 10 mM DTT and heated at 56 °C for 30 min. The solutions were replaced with 200 μL of 100 mM TEAB buffer containing 55 mM iodoacetamide and incubated for 45 min in the dark. The gel pieces were washed with 100 mM TEAB buffer and dehydrated in acetonitrile. This step was repeated once. The gels were reswelled in 100 mM TEAB buffer containing 10 ng/μL Sequencing Grade Trypsin (Promega) and incubated overnight at 37 °C. To the gels, 50 μL of an extraction solution (50% acetonitrile, 0.1% TFA) was added. After 10 min incubation, the supernatants were collected to new tubes, and this process was repeated twice. Extracted peptides were dried in vacuo and reconstituted with 25 μL of 100 mM TEAB buffer. 6-plex TMT reagents (Thermo Fisher Scientific) were used for quantitative MS analysis of samples treated with **11**, **11** with PU-H71, or DMSO (light tag for **11**, medium tag for **11** with PU-H71, and heavy tag for DMSO) according to manufacturer's instruction. Briefly, the digested peptides were allowed to react with TMT reagents for 1 h at room temperature, and the reactions were quenched by addition of 5% hydroxylamine. Three samples were mixed in a 1:1:1 ratio and dried. The peptides were dissolved in 200 μL of 5% acetonitrile / 0.1% TFA mixture and purified by a GL-Tip SDB (GL Sciences) according to the manufacturer's instruction. Briefly, the tip was washed with 5% acetonitrile / 0.1% TFA mixture, and peptides were eluted by 80% acetonitrile / 0.1% TFA. After removing the solvent, the residue was dissolved in 43 μL of a 5% acetonitrile / 0.1% TFA mixture. NanoLC–MS/MS analyses were carried out on a Q-Exactive mass spectrometer (Thermo Fisher Scientific) equipped with an Ultimate 3000 nanoLC pump (AMR). Samples were automatically injected using PAL system into a peptide L-trap column OSD (5 μm) attached to an injector valve for desalinating and concentrating peptides. After washing the trap with MS-grade water containing 0.1% TFA and 2% acetonitrile, the peptides were loaded into a nano HPLC capillary column (C18 packed with the gel particle size of 3 μm, 0.1 × 150 mm, Nikkyo Technos, Tokyo Japan). The injection volume was 5 μL and the flow rate was 500 nL/min. The mobile phases consisted of (A) 0.1% acetic acid and (B) 0.1% acetic acid and 80% acetonitrile. A two-step linear gradient of 5–45% B in 60 min, 45–95% B in 1 min, 95% B for 20 min was performed. Spray voltage, the mass scan ranges, and the normalised collision energy were set to be 2000 V, $m/z$ 120–1800, and 30, respectively. Top five precursor ions were selected in each MS scan for subsequent MS/MS scans. A lock mass function was used to obtain constant mass accuracy during gradient. The MS data was analysed by Proteome Discoverer 2.2 (Thermo Fisher Scientific). Searches were performed using Sequest HT (Thermo Fisher Scientific) against UniprotKB/Swiss-Prot release 2017-07-05 with a precursor mass tolerance of 10 ppm, a fragment ion mass tolerance of 0.02 Da. Trypsin specificity allowed for up to two miscleavages. TMT labelling on lysine residues and peptide N-termini, and cysteine carbamidomethylation were set as static modifications. Methionine oxidation was set as a dynamic modification. A reversed decoy database search was carried out to set FDRs of less than 0.01 both at peptide and protein levels. Proteins detected with a minimum of three peptides at least twice were selected as identified proteins in three replicates, in which keratin proteins were removed. Protein quantification was performed by averaging relative peak intensities of the TMT reporter signals across all quantified peptides.

**Identification of labelling site of Hsp90 labelled with 11**. Before labelling, HeLa cells ($1.2 × 10^6$ cells) were seeded on 10 cm dish and incubated in DMEM supplemented with 10% FBS for 48 h at 37 °C under 5% CO$_2$. After washing twice with PBS, the cells were incubated in DMEM (HEPES-modified, FBS free) containing LDNASA **11** (0.5 μM) for 3 h. The cells were washed with PBS, and lysed by three successive freeze and thaw steps in Felts buffer (20 mM HEPES, 50 mM KCl, 5 mM MgCl$_2$, 0.01% (w/v) NP-40, freshly prepared 20 mM Na$_2$MoO$_4$, pH 7.2–7.3) containing 1% protease inhibitor cocktail. The lysed sample was centrifuged (15 200 × $g$, 10 min at 4 °C), and the protein concentration of supernatant was analysed by BCA assay. The supernatant was mixed with Protein G Sepharose 4 Fast Flow (GE Healthcare) and rotated at 4 °C for 1 h. After removal of the beads, the remaining supernatant was mixed with anti-fluorescein antibody (Abcam, ab19491, 1:240) and rotated at 4 °C for 1 h, followed by addition of fresh Protein G Sepharose 4 Fast Flow and further incubation at 4 °C for 1 h. The beads were washed five times with RIPA buffer and once with PBS. Proteins were eluted from

the beads by addition of 2 × sample buffer containing 100 mM DTT and boiling at 95 °C for 5 min. The samples were resolved by 7.5% SDS-PAGE, and the fluorescent band corresponding to fluorescein-modified Hsp90 was excised from in-gel fluorescence image. The excised gel was subjected to in-gel digestion using MS grade Trypsin (Thermo Fisher Scientific). The peptide fragments were analysed by nanoflow reverse liquid chromatography followed by tandem MS, using a LTQ Orbitrap XL hybrid mass spectrometer as described above. Searches were performed using the SEQUEST HT against the latest uniprot database for Hsp90 alpha (HS90A_HUMAN, P07900) and Hsp90 beta (HS90B_HUMAN, P08238).

**In vitro labelling of N-terminal domain of Hsp90α**. Recombinant Hsp90α N-terminal domain was obtained as described in Supplementary Methods. The solution of Hsp90α N-terminal domain (6.4 μM) in PBS was incubated with **12** (10 μM) in PBS at 37 °C. Aliquots at different time points were taken and then desalted using a Ziptip-C4. The modification yields were determined by MALDI-TOF MS (matrix: sinapic acid)

**Peptide mapping of N-terminal domain of Hsp90α**. The **12**-modified N-terminal domain of Hsp90α was purified by size-exclusion chromatography as described above. The purified protein was denatured by urea (at a final concentration of 2 M), followed by tryptic digestion (Trypsin/substrate ratio = 1/10 (w/w)) at 37 °C for 18 h. The digested peptides were separated by RP-HPLC and characterised by MALDI-TOF MS and MS/MS (matrix: CHCA).

**Cell viability assay**. SKBR3 cells ($6 × 10^4$ cells) were seeded on a 12 well plate (Corning) and incubated in McCoy's 5A supplemented with 10% FBS for 48 h at 37 °C under 5% CO$_2$. After washing twice with HEPES-modified McCoy's 5A (FBS-free) medium, the cells were incubated in the medium containing PU-H71 or **12** (0.3–10 μM) for 3 h at 37 °C. The cells were washed twice with McCoy's 5A supplemented with 10% FBS and further incubated for 69 h. As a control experiment, the cells were treated with PU-H71 for 72 h. The cell viability was assessed using Cell Counting Kit-8 (Dojin). The absorbance of each well was measured at 450 nm with infinite M200 (TECAN).

**Western blotting of Hsp90 client proteins**. SKBR3 cells ($2 × 10^5$ cells) were seeded on a 12 well plate and incubated in McCoy's 5A supplemented with 10% FBS for 48 h at 37 °C under 5% CO$_2$. After washing twice with HEPES-modified McCoy's 5A (FBS-free) medium, the cells were incubated in the medium containing PU-H71 (10 μM) or **12** (1–10 μM) for 3 h at 37 °C. The cells were washed twice with McCoy's 5A supplemented with 10% FBS and further incubated for 21 h. The cells were washed with PBS, and lysed with RIPA buffer containing 1% protease inhibitor cocktail. The lysed samples were centrifuged (15 200 × $g$, 10 min at 4 °C). The protein concentrations of supernatant were analysed by BCA assay, and the normalised lysates were mixed with 1/4 volume of 5 × sample buffer containing 250 mM DTT and vortexed for 1 h at room temperature. The samples were subjected to Western blotting analysis using anti-HER2 (CST, #2242, 1:1000), cRaf (CST, #53745, 1:1000), pAkt (CST, #4060, 1:2000), Akt (CST, #4691, 1:1000), and beta actin (Abcam, ab8226, 1:1000) antibodies.

**Data availability**. The data that support the findings of this study are available from the corresponding authors on reasonable request.

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

## Acknowledgements

We thank Karin Nishimura and Eriko Kusaka (Kyoto University) for experimental support of MS and NMR measurements. This work was supported by Grant-in-Aid for Young Scientists (B) (15K17884) and The Kyoto University Foundation to T.T., and the Japan Science and Technology Agency (JST) Core Research for Evolutional Science and Technology (CREST) to I.H. This work was also supported by a Grant-in-Aid for Scientific Research on Innovative Areas Chemistry for Multimolecular Crowding Biosystems (JSPS KAKENHI Grant no. 17H06348).

## Author contributions

T. Tamura and I.H. conceived and designed the project. T. Tamura., T.U., T.G., T. Tsukidate, Y.S. and Y.N. synthesized the labelling reagents. T. Tamura and T.G. carried out the kinetics experiments and cell-based protein labelling. T.U. conducted labelling and covalent inhibition of Hsp90. A.F. identified the labelling site of Hsp90. T. Tamura., T.U., T.G. and I.H. analyzed the experimental data. The manuscript was written by T. Tamura and I.H., and edited by all the co-authors.

## Additional information

**Competing interests:** The authors declare no competing interests.

