## [Peer Review File · Nature Communications]

Reviewers' comments:

Reviewer #1 (Remarks to the Author):

The research manuscript "Rapid labelling and covalent inhibition of intracellular native proteins using ligand-directed N-acyl-N-alkyl sulfonamide" by Tamura et al. introduces N-acyl-N-alkyl sulfonamide compounds fused to ligands that label proteins near the ligand's binding site.

In principle, the concept of ligand-directed chemistry is not novel, nevertheless the authors describe in great detail the exceptional speed of newly developed chemistry in the context of ligand-directed reactions. One should note that the downside of the fast reaction speed seems to be a loss of specificity for the target.

The authors do put significant work into characterizing this unspecific labelling, for example in the case of the lysate-spiking experiment shown in figure 4a and the supplementary figure S10. In this case when applying 10 μM of their labelling reagent to cell lysate containing 1 μM of their target protein, 14% of the labelling is unspecific. This is quite a high proportion of unspecific labelling, especially considering the working concentration that the authors use in later experiments is 10 μM . The authors also determine the labelling sites for the FKBP12, eDHFR and HSP-90 proteins to make sure labelling occurs at the intended positions. The authors also demonstrate, to the detriment of their technique, that a very high affinity ligand is required for successful application (figure 4a). In this sense, the authors are quite thorough in characterizing the downsides of their newly developed chemistry.

While the unspecific labelling demonstrated here may detract potential users from applying this chemistry, it could have potential for applications in which speed is much more important than specificity. The authors do provide an interesting study into the reaction kinetics of this chemistry in comparison to related methods, and an application example with the development of an irreversible inhibitor for the chaperone protein Hsp90.

Despite the limitations of the technique, the work shown here is a valuable and interesting addition to the field. For this reason, I recommend this article to be accepted for publication, although the authors should consider also mentioning these limitations more clearly in the abstract and discussion sections.

Reviewer #2 (Remarks to the Author):

The authors have addressed the vast majority of my concerns, although I believe that some "hand waving" was invoked. Nonetheless, the resulting manuscript is much stronger. However, one issue that remains is in regard to the "prolonged residential time of NASA group close to" the Lys. Based on what the author has stated, I believe this is probably incorrect, as increased residence time is likely to result from covalent modification, not increased residence time leads to covalent modification. This statement should be scrutinized more carefully.

p24, line 6: "...NASA group is capable to label the..." change to "NASA group is capable of labeling the..."

Figure 4a legend: "...signals of lane 6 and 7 may suggest many of the unspecific..." change to "...signals of lane 6 and 7 may suggest unspecific"... "were less in lane 1-5" add "s" to lane to construct lanes

Reviewer #3 (Remarks to the Author):

I recommend this article to be accepted for publication on Nature Communications, since the manuscript showed us a great work after revisions.

Point-by-point response to the referees' comments

Reviewers' comments:

Reviewer #1 (Remarks to the Author):

The research manuscript “Rapid labelling and covalent inhibition of intracellular native proteins using ligand-directed N-acyl-N-alkyl sulfonamide” by Tamura et al. introduces N-acyl-N-alkyl sulfonamide compounds fused to ligands that label proteins near the ligand's binding site.

In principle, the concept of ligand-directed chemistry is not novel, nevertheless the authors describe in great detail the exceptional speed of newly developed chemistry in the context of ligand-directed reactions. One should note that the downside of the fast reaction speed seems to be a loss of specificity for the target.

The authors do put significant work into characterizing this unspecific labelling, for example in the case of the lysate-spiking experiment shown in figure 4a and the supplementary figure S10. In this case when applying 10 μM of their labelling reagent to cell lysate containing 1 μM of their target protein, 14% of the labelling is unspecific. This is quite a high proportion of unspecific labelling, especially considering the working concentration that the authors use in later experiments is 10 μM .

Thank you for your careful comment. We would like to note that the proportion of unspecific reaction in the lysate-spiking experiment is not always the same in any situation. For example, we clearly showed that the unspecific labelling did not increase even with 10 μM of the reagent in the case of Hsp90 labelling (Figure S12). This suggests that the concentration of the reagent causing unspecific labelling varies depending on the target protein, the reagent properties (e.g. affinity, hydrophobicity), and the reaction contexts. Nevertheless, as this reviewer commented, we agree that 14% of unspecific labelling in the lysate experiment is not negligible. Thus, we modified the manuscript as follows:

[Original manuscript, p 15, line 14]

Although several bands due to unspecific labelling appeared in the ratio of reagent-to-protein greater than one, these were not so significant amount compared with the labelling band of FKBP12 (the ratio of unspecific to specific labelling is less than 0.15).

[Revised manuscript, p 15, line 14]

Although several bands due to unspecific labelling appeared in the ratio of reagent-to-protein greater than one, the labelling band of FKBP12 was predominant (the ratio of unspecific to specific labelling is less than 0.15).

The authors also determine the labelling sites for the FKBP12, eDHFR and HSP-90 proteins to make sure labelling occurs at the intended positions. The authors also demonstrate, to the detriment of their technique, that a very high affinity ligand is required for successful application (figure 4a). In this sense, the authors are quite thorough in characterizing the downsides of their newly developed chemistry.

While the unspecific labelling demonstrated here may detract potential users from applying this chemistry, it could have potential for applications in which speed is much more important than specificity. The authors do provide an interesting study into the reaction kinetics of this chemistry in comparison to related methods, and an application example with the development of an irreversible inhibitor for the chaperone protein Hsp90.

Despite the limitations of the technique, the work shown here is a valuable and interesting addition to the field. For this reason, I recommend this article to be accepted for publication, although the authors should consider also mentioning these limitations more clearly in the abstract and discussion sections.

According to the reviewer's comment, we modified the abstract and added a new description about the limitations of our method in the conclusion section as follows:

[Abstract]

Selective modification of native proteins in live cells is one of the central challenges in recent chemical biology. As a unique bioorthogonal approach, ligand-directed chemistry recently

emerged, but the slow kinetics limits its scope. Here we have successfully overcome this obstacle using *N*-acyl-*N*-alkyl sulfonamide (NASA) as a new reactive group. Quantitative kinetic analyses revealed that ligand-directed NASA allowed for rapid modification of a lysine residue proximal to the ligand binding site of a target protein, with a rate constant of $\sim 10^4 \text{ M}^{-1}\text{s}^{-1}$, comparable to the fastest bioorthogonal chemistry. Despite some off-target reactions, this method can selectively label both intracellular and membrane-bound endogenous proteins. Moreover, the NASA-mediated protein labelling enables the rational design of a lysine-targeted covalent inhibitor that shows durable suppression of the activity of Hsp90 in cancer cells. Ligand-directed NASA chemistry provides new possibilities to extend the covalent inhibition approach that is currently being reassessed in drug discovery.

[Conclusion]

In summary, we have developed LDNASA chemistry that shows rapid reaction kinetics with sufficient target selectivity and bioorthogonality under native live cell conditions. The second-order rate constant was comparable to that of the fastest bioorthogonal protein modification methods. We also found that the NASA reactive group is capable of labelling the ϵ -amino group of non-catalytic Lys residues. It is generally considered that $\sim 99.9\%$ of the Lys residues on protein surface are protonated, which suppresses their nucleophilicities (reactivities) under physiological pH.²⁵ However, our data strongly suggest that the chemical labelling of the Lys residues can proceed by the close proximity of the NASA reactive group to the ϵ -amino group of Lys. While the ligand-directed chemistry requires an appropriate ligand and may suffer to some extent from the unspecific labelling in biological crude environments, the present study indicated that this limitation can be addressed by careful design of labelling reagents (use of a ligand with a K_d value of sub μM order) and thorough optimization of reaction conditions. Moreover, we demonstrated that the mass spectroscopy-based analysis is quite powerful for the proteome-wide characterization of the possible off-target proteins in this method. These results provide deep insights and important guidelines for practical use of not only LDNASA chemistry but also other proximity-driven protein labelling in living systems.

Reviewer #2 (Remarks to the Author):

The authors have addressed the vast majority of my concerns, although I believe that some "hand waving" was invoked. Nonetheless, the resulting manuscript is much stronger. However, one issue that remains is in regard to the "prolonged residential time of NASA group close to" the Lys. Based on what the author has stated, I believe this is probably incorrect, as increased residence time is likely to result from covalent modification, not increased residence time leads to covalent modification. This statement should be scrutinized more carefully.

According to this comment, we modified the manuscript as follows:

[Original manuscript, p.24, line 9]

However, our data strongly suggest that the chemical labelling of the Lys residues can proceed by the prolonged residential time of a reactive group close to such Lys (i.e. proximity effect).

[Revised manuscript, p.24, line 8]

However, our data strongly suggest that the chemical labelling of the Lys residues can proceed by the close proximity of the NASA reactive group to the ϵ -amino group of Lys.

p24, line 6: "...NASA group is capable to label the..." change to "NASA group is capable of labeling the..."

We appreciate the reviewer's indication. This grammatical mistake has been corrected in the revised manuscript.

Figure 4a legend: "...signals of lane 6 and 7 may suggest many of the unspecific..." change to "...signals of lane 6 and 7 may suggest unspecific" ... "were less in lane 1-5" add "s" to lane to construct lanes

According to this comment, we modified the manuscript. Please see Figure 4a legend.

Reviewer #3 (Remarks to the Author):

I recommend this article to be accepted for publication on Nature Communications, since the manuscript showed us a great work after revisions.

We thank the reviewer for the positive comment about our revised manuscript.